# Rheumatoid Arthritis and Reactive Oxygen Species: A Review

**Naoki Kondo [1,*], Tomotake Kanai [1] and Masayasu Okada [2]**

[1] Division of Orthopedic Surgery, Department of Regenerative and Transplant Medicine, Graduate School of Medical and Dental Sciences, Niigata University, Niigata 951-8510, Japan

[2] Department of Neurosurgery, Brain Research Institute, Niigata University, Niigata 951-8510, Japan

[*] Correspondence: naokikondo1214@gmail.com; Tel.: +81-25-227-2272

**Abstract:** Rheumatoid arthritis (RA) is a chronic, systemic inflammatory disease that causes progressive joint damage and can lead to lifelong disability. Numerous studies support the hypothesis that reactive oxygen species (ROS) are associated with RA pathogenesis. Recent advances have clarified the anti-inflammatory effect of antioxidants and their roles in RA alleviation. In addition, several important signaling pathway components, such as nuclear factor kappa B, activator-protein-1, nuclear factor (erythroid-derived 2)-like 2/kelch-like associated protein, signal transducer and activator of transcription 3, and mitogen-activated protein kinases, including c-Jun N-terminal kinase, have been identified to be associated with RA. In this paper, we outline the ROS generation process and relevant oxidative markers, thereby providing evidence of the association between oxidative stress and RA pathogenesis. Furthermore, we describe various therapeutic targets in several prominent signaling pathways for improving RA disease activity and its hyper oxidative state. Finally, we reviewed natural foods, phytochemicals, chemical compounds with antioxidant properties and the association of microbiota with RA pathogenesis.

**Keywords:** rheumatoid arthritis; reactive oxygen species; pathogenesis; therapeutic targets; antioxidants

## 1. Introduction

Rheumatoid arthritis (RA) is a chronic, systemic inflammatory disease that causes progressive joint damage and can lead to lifelong disability [1]. RA is characterized by synovial inflammatory cell infiltration, synovial hyperplasia, angiogenesis, and cartilage damage, which in turn can lead to bone degradation [2]. Recent data have demonstrated that bone and cartilage degradation in RA are due to an increase in metalloproteinases (MMPs) and serin proteases [3]. Many studies have reported that circulating neutrophils show an aberrant, activated phenotype in RA, characterized by delayed apoptosis and the increased production of reactive oxygen species (ROS) and cytokines, resulting in bone and joint damage [4–7].

Numerous studies have reported an association between RA pathogenesis and ROS [8–10]. For example, circulating neutrophils in RA patients can generate superoxide anions ($O^{2-}$), unlike those in healthy controls [11]. Moreover, the levels of catalase and ceruloplasmin were remarkably elevated in the synovial fluid of RA patients compared to those of controls, suggesting that antioxidant activity was enhanced in RA pathogenesis in response to inflammation (Figure 1) [12].

The overproduction of nitric oxide (NO) contributes to the pathogenesis of chronic arthritis [13]. In collagen-induced rodent arthritis models, increased levels of nitrite/nitrate in the plasma [14,15] and synovial fluid [16] and a high expression of inducible nitric oxide synthase (NOS) in proliferating synovium [14] and chondrocytes [17] have been reported. Increased circulating levels of nitrate/nitrite have been detected in arthritis patients [18], and the synovial tissues of RA patients were characterized by high iNOS expression [19,20] and enhanced NO production [14].

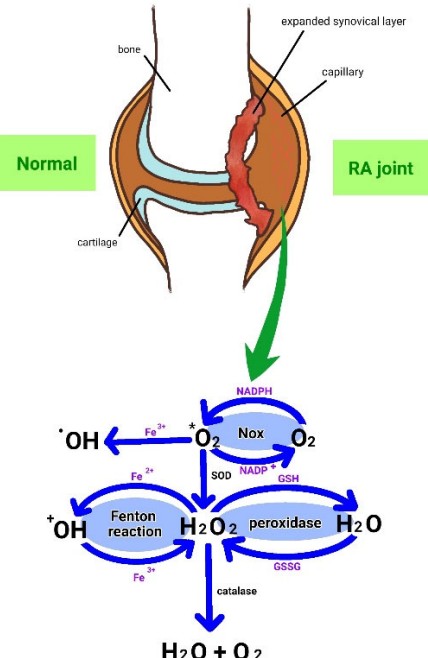

**Figure 1.** Bone and joint damage and the generation of reactive oxygen species in an inflamed joint affected by rheumatoid arthritis. Abbreviations: RA, rheumatoid arthritis; GSH, reduced glutathione; GSSH, oxidized glutathione; NADPH, nicotine amide adenine dinucleotide phosphate; SOD, superoxide dismutase.

In this review, the relationship between ROS and RA is summarized. ROS are briefly described, and the association between antioxidants and RA treatment is reviewed by focusing on neutrophils and the autophagy of synovial fibroblasts in RA. Redox signaling in RA is also discussed. Finally, natural food or antioxidants with the potential for improving RA disease severity and novel therapeutic targets are enumerated.

## 2. Oxidative Stress and Markers

There are two methods for measuring oxidative DNA damage. Steady-state damage can be measured when DNA is isolated from human cells or tissues and analyzed for base damage products. Several DNA base damage products, such as nucleoside 8-hydroxy-deoxyguanosine (8-OH-dG), 8-hydroxy-adenine, and 7-methyl-8-hydroxyguanine, are excreted in human urine [21–23]. The most used marker is 8-OH-dG, which is usually measured using high-performance liquid chromatography. The level of 8-OH-dG in urine is probably not influenced by the diet, as nucleotides are not absorbed from the gut. In RA patients, 8-OH-dG level was significantly decreased by methotrexate or TNF-inhibitors, such as infliximab and etanercept [24–26].

Biologics are widely used in the clinical setting in patients with RA; however, side effects are a concern. If other targets, such as antioxidants or ROS-inhibitory or antioxidant-promoting agents, are induced, the clinical symptoms of RA, the number of tender joints, arthralgias, joint swellings, and osteoarthritic joint disorders, are reduced. Physical activity, such as knee flexion, can also be improved. Commonly used oxidative stress markers include 8-hydroxy-2-deoxyguanosine (8-OHdG), thiobarbituric acid reactive substances, malondialdehyde (MDA), isoprostane (IsoPs) and its metabolites, and allantoin [27,28] (Table 1), as well as advanced glycation end product (AGE) [29]. IsoPs is the isomer of prostaglandin, produced from polyunsaturated fatty acids involved in the peroxidation of phospholipid membranes by free radical or oxidative stress [30]. Allantoin is the final metabolite in the non-enzymatic oxidation of uric acid. Although a small number of cases were reported, allantoin levels were significantly higher in RA patients than in healthy

subjects. Uric acid acts as a scavenger of free radicals, and allantoin increases reactivity, suggesting the involvement of free radicals in RA pathophysiology [31].

NADPH oxidase is the most important generator of ROS in the vasculature [32,33]. It comprises membrane-bound gp91 (phox) and gp22(phox) and cellular solute subunits, such as p47(phox), p67 (phox), and the small GTPase Rac. Neutrophil cytosolic factor-1 (NCF-1), also known as p47 phox, which is an essential subunit of NADPH oxidase 2 (NOX2), an enzyme that promotes oxidative stress. The activation of NOX2 first occurs in the cytoplasm. The extent of oxidative stress is derived from the production of superoxides, which is called oxidative disruption oxidative burst. Genetic polymorphisms in the genes coding for the Nox2 complex have become a hot topic in human immune diseases. ROS derived from the NCF1 and NOX2 complex are important regulators of rheumatoid arthritis, multiple sclerosis, psoriasis, psoriatic arthritis, gout, lupus, and other chronic inflammatory diseases [34]. Table 1 summarizes the oxidative stress markers that are available (Table 1) [24–26,35–43].

**Table 1.** Markers of oxidative stress.

| Process | Markers | Samples | Explanations | References |
|---|---|---|---|---|
| Nucleic acid oxidation | 8-hydroxy-2′-deoxyguanosine | Urine, serum, tissue | Guanine receives oxidative stress and 8-OHdG is produced by oxidation of carbon and excreted in urine. | [24–26,35] |
| Lipid peroxidation | 4-hydroxynonenal (CloneHNEJ-2) | Tissue | Representative oxidative stress product generated at the late stage of lipid peroxidation | [36,37] |
| | 15-Isoprostane F2t | Urine, serum, tissue | Oxidized phospholipids by free radicals | [38] |
| | Malonedialdehyde (MDA) | Urine, serum, tissue | MDA derives from polyunsaturated lipid acid and reacts to thiobarbitulic acid (TBA) and is detected spectroscopically. | [39] |
| | Lipid peroxidase (LPO) | Serum, tissue | LPO is detected by the methylene blue/hemoglobin method. | [40] |
| | Oxidized-LDL | Serum, tissue | Generated free radicals oxidizes lipids in LDL and oxidized LDL is formed. | [41] |
| Glucose oxidation | Pentosidine | Urine, serum | Main advanced glycation end products (AGEs) Glycated albumin is a glycation product of albumin. | [24] |
| | Dithyrosine | Urine, serum, tissue | Dityrosine is a tyrosine dimer that is formed by the oxidation of tyrosine. | [42] |
| NO stress | Nitrotyrosine (NT) | Serum, tissue | NT is formed by the nitration of protein-bound and free tyrosine residues by reactive peroxinitrite molecules. | [43] |

## 3. Neutrophils in RA and ROS

Neutrophils are differentiated cells. In the absence of inflammation, they circulate in the blood for 24–48 h until they return to the bone marrow, leading to apoptosis [44]. In the presence of inflammation, neutrophil apoptosis is delayed by inflammatory cytokines, such as tumor necrosis factor-alpha (TNF-alpha) and granulocyte macrophage colony-stimulating factor (GM-CSF).

The hypoxic environment in RA synovial joints also plays a key role in delaying neutrophil apoptosis by increasing MCL1 expression [45]. Hypoxia can also delay apoptosis via the stabilization of hypoxia-inducible factor-1-alpha (HIF1-$\alpha$) and the activation of nuclear factor-kappa B (NF-$\kappa$B) [46]. Furthermore, hypoxia regulates neutrophil retention at the sites of inflammation, thereby prolonging inflammation [47].

## 4. Autophagy and ROS in RA-SF

Autophagy is involved in the transformation of RA-SF, and microRNA is one of these. MicroRNA (miRNA) is a small non-coding RNA consisting of 18 to 25 nucleotides in length [48]. By downregulating the mRNA translocation of downstream target genes, miRNA suppresses gene expression. miRNAs are essential for cell proliferation, apoptosis, oxidative stress, and immune response. In particular, miR-19 [49], miR-21 [50], MiR-27a [51], and MiR-29a have been reported to be involved in RA pathogenesis. In addition, MiR-650 was significantly less expressed in RASF than in normal cells, whereas AKT2 was highly expressed. The downregulation of MiR650 or the upregulation of AKT2 increased RASF proliferation, migration, and erosion and suppressed apoptosis [52]. miR-126 was associated with PI3KR2 as a target gene, and its overexpression suppressed P13KR2 expression, promoted RASF proliferation, and suppressed apoptosis; thus, miR-126 is a candidate for the predictive biomarker of RA [53]. miR-218-5p was highly expressed in RASFs. Its inhibition severely suppressed the production of oxidative stress and promoted SOD [54]. Collectively, the knockdown of miR-218-5p increased KLF9 expression through the downregulation of the JAK2/STAT3 signaling pathway, suggesting that it may be a potential therapeutic target for controlling RASF growth, apoptosis, and oxidative stress [54].

The myeloid-specific deletion of the gene encoding IRE1$\alpha$ protected mice from inflammatory arthritis, and the IRE1$\alpha$-specific inhibitor 4u8c attenuated joint inflammation in mice [55]. Recently, oxidative stress was reported to be associated with autophagy/ER stress in the pathogenesis of RA [34,56,57]. Thus, the IRE1/JNK pathway might be a therapeutic target for regulating oxidative stress in RA.

## 5. Main Transcriptional Factors Associated with ROS

### 5.1. NF-$\kappa$B

NF-$\kappa$B was the first eukaryotic transcription factor shown to respond directly to oxidative stress [58]. It plays a key role in the regulation of numerous genes involved in immune and inflammatory processes [59]. TNF$\alpha$, IL-1, phorbol ester, lipopolysaccharide, and UV radiation potently activate NF-$\kappa$B in intact cells. The $H_2O_2$ exposure of several types of cells rapidly induced NF-$\kappa$B activation, indicating that $H_2O_2$ might be a mediator of prooxidant-induced NF-$\kappa$B activation.

### 5.2. AP-1

Activator protein-1 (AP-1) is a transcription factor for regulating collagen genes, TNF$\alpha$, IL8, IL9, IL3, IFN$\gamma$, adhesion molecules related to the formation of atherosclerotic plaques, and genes involved in the cell division cycle [60]. AP-1 activity is induced in response to certain metals in the presence of $H_2O_2$ and several cytokines and other physical and chemical stressors.

## 6. Redox Signaling

Redox signaling refers to a regulatory process in which the signal is transduced through redox reactions [61]. NF-$\kappa$B, hypoxia-inducible factor-1 (HIF-1), AP-1, and Nrf2 are redox-sensitive transcriptional factors. They are closely involved in the pathogenesis of RA. NF-$\kappa$B is crucial for the maturation of immune cells and the production of TNF$\alpha$ and MMPs. TNF$\alpha$ and MMPs aggravate RA. HIF-1 is induced by inflammatory cytokines and needed for angiogenesis and pannus formation in RA. AP-1 and IL-1 beta affect the gene expression and activity of each other, which results in an orchestrated cross-talk. AP-1 also regulates MMP production and synovial hyperplasia in RA [61].

Nrf2 triggers the first line of homeostatic response against endogenous deviations in redox metabolism, proteostasis, and inflammation [62]. Nrf2 deficiency worsened disease activity in experimental arthritis models, whereas its activation exhibited immunoregulatory and anti-inflammatory effects. Thus, the pharmacologic regulation of Nrf2 has gained increasing interest as a strategy to target ROS [63].

In THP-1 monocytes/macrophage cells, light-emitting diode irradiation at 630 nm significantly reduced ROS levels and inhibited the expression of TNF$\alpha$ and IL-1$\beta$ mRNA. Lastly, the level of phosphorylated NF-$\kappa$B was significantly reduced, whereas that of its inhibitor, Nrf2, was slightly upregulated [64].

## 7. New Therapeutic Targets

Poly-(ADP-ribose) polymerase-1 (PARP-1) is a member of the PARP enzyme family, consisting of PARP-1 and several other additional poly-(ADP-ribosylating) enzymes [65]. Peroxynitrite-dependent cell necrosis is partially mediated by a complex process involving DNA damage and the activation of the DNA repair enzyme PARP-1 [66]. PARP-1 detects and transmits the signals of DNA strand breaks induced by various genotoxic insults and oxidants (hydrogen peroxide and peroxynitrite) and free radicals (mainly carbonate or hydroxyradicals) [67,68]. A significant increase in DNA strand breaks in peripheral mononuclear cells was observed in RA patients compared with that in healthy subjects [69,70]. PARP was highly expressed in the joint tissues of collagen-induced arthritis rodent models. Following peroxynitrite formation blockage by selective iNOS inhibitors or the suppression of genetic iNOS, PARP activation was blocked [14]. Several PARP-1 inhibitors, such as nicotinamide [71], 5-inodo-6-amino-1,2-benzopyrone [72], and PJ-34 [73], were previously used.

Scavenging NO might be an alternative strategy for treating inflammatory disorders. Yeo et al. [74] developed a NO-responsive macro-sized hydrogel by incorporating a NO-cleavable cross-linker. The NO-scavenging nanosized hydrogel (NO-Scv gel) reduced inflammation levels by scavenging NO in vitro. Furthermore, the NO-Scv gel suppressed RA onset in a mouse RA model compared with the dexamethasone treatment.

Mateen et al. [75] identified the association of inflammatory cytokines with 25-hydroxy vitamin D (25-OH-D) and ROS in RA patients. The level of 25-OH-D in RA patients was $11.07 \pm 4.81$ ng/mL, which was significantly reduced compared to that in healthy controls ($37.88 \pm 9.78$ ng/mL). Another study using the DCF-DA method showed that the ROS levels in RA patients were significantly increased compared to those of healthy controls [76].

## 8. Natural Foods, Phytochemicals, and Chemical Compounds with Antioxidant Properties and the Association of Microbiota with RA Pathogenesis

Once organisms are exposed to free radicals, a series of defense mechanisms are activated, one of which is represented by antioxidants. The most prevalent enzymatic antioxidants are superoxide dismutase (SOD), glutathione peroxidase (GPx), and catalase (CAT). Nonenzymatic antioxidants, such as ascorbic acid (vitamin C), $\alpha$-tocophenol (vitamin E), glutathione (GSH), carotenoids, flavonoids, and other antioxidants, also play an important role.

Various interventions have been performed for the dietary pattern disease activity of rheumatoid arthritis [77,78].

Private diet eliminating meat, gluten, and lactose for 3 months in 40 patients with RA significantly decreased in pain, DAS28 scores, CRP level, and the overall state of physical and mental health [79].

A dietary intake of vitamins C and E, zinc, magnesium, copper, and selenium was introduced in 87 RA patients. Of these dietary components, vitamin C intake was related to decreased IL-1b, zinc intake was related to decreased IL-2, and magnesium intake was related to decreased levels of IL-1b and IL-2. In addition, vitamin E and copper intake increased catalase (an enzyme largely involved with anti-inflammatory pathways) expression [80].

The dietary inflammatory index (DII) was assessed for patients with RA and control (case–control study. The mean DII score was higher in the RA patients compared with control cases (0.66 vs. −0.58, $p = 0.002$). Higher DII scores were significantly correlated with higher CRP, TNF-alpha, DAS-28 scores, and the number of tender joints [81].

The evaluation of diet quality is also important. Participants with lower diet quality showed significantly higher pain and ESR scores [82]. The anti-inflammatory diet in rheumatoid arthritis diet (ADIRA) for 44 patients with RA demonstrated a significant decrease ($p = 0.012$) in DAS28-ESR as a randomized controlled test [83]. In addition, poor diet quality, as defined by Swedish National Food agency (diet with a low intake of fish, shellfish, whole grain, fruit, and vegetables and a high intake of sausages and sweets), was associated with higher CRP ($p = 0.044$) and ESR ($p = 0.002$) levels in patients with RA [84].

The Mediterranean diet (MD) is effective for the decrease in inflammation for patients with RA. Patients with high adherence to the MD showed a significantly lower CRP ($p = 0.037$) and DAS28 ($p = 0.034$) than the 40 patients with low or moderate adherence to MD. A healthier gut microbiota status was detected in the high adherence group [85].

Diurnal fasting for 1 month is also effective. A significant decrease in visual analogue pain scores, tender and swollen joints ($p = 0.02$), DAS-28 ($p = 0.003$), and ESR was observed [86]. RA patients who continued the diurnal fasting of Ramadan demonstrated a significant improvement in DAS28-CRP ($p = 0.001$) and DAS28-ESR ($p < 0.001$) compared with patients who did not participate in the fasting [87].

In anthropometric findings, body mass index (BMI) in patients with RA was correlated with CRP (r = 0.36, $p < 0.01$) and ESR (r = 0.31, $p < 0.01$). Asymmetric dimethylarginine (ADMA) is a naturally occurring chemical found in blood plasma. ADMA was associated with increased BMI and disease RA activity. A higher intake of protein was correlated to higher CRP and ESR [88].

Supplementation with high fiber 30 g bars daily for 15 days and 30 days were administered to 10 healthy controls and 29 patients with RA. Increased anti-inflammatory short-chain fatty acids was detected ($p < 0.001$). Proarthritic cytokines concentrations, such as MCP-1, IL-18, and IL-33, were decreased and the Firmicutes-to-Bacteroides ratio, one of markers of gut microbiota, were decreased ($p < 0.05$) [89].

Naïve human CD4+ T cells were cultured in 10, 20, 40, and 60 mM NaCl solution for 3 days. NaCl aggravated arthritis by affecting Th17 differentiation [90]. A low sodium diet, as defined as less than 5 μg/day for 3 weeks, significantly reduced the serum levels of transforming growth factor-beta (TGFβ) and IL-9 [91].

The efficacy of flavonoids, PUFAs, and probiotics in the disease activity of RA has also been reported.

Black barberry extract (1000 mg/day for 12 weeks) intervention significantly decreased IL-17 levels and increased IL-10 [92]. The trial of Brazilian propolis (508.5 mg daily for 24 weeks) did not show a significant difference in DAS28-ESR, CRP, simplified disease activity index, or clinical disease activity index [93]. Cinnamon powder (500 mg daily for 8 weeks) showed a significant decrease in the serum levels of CRP and TNF-alpha. It also showed a significant decrease in DAS-28, visual analogue scale, and the tender and swollen joints count ($p < 0.001$) [94]. Cinnamaldehyde and eugenol on peripheral blood mononuclear cells showed significant dose-dependent decreases in TNF-alpha and IL-6 ($p < 0.05$), ameliorated reactive oxygen species formation, biomolecular oxidation, and antioxidant defense response ($p < 0.05$) [95].

Curcumin nanomicelle (120 mg daily for 12 weeks) demonstrated no significant decrease in the DAS-28, tender joint count, swollen joint count, and ESR after intervention [96].

However, increased doses of curcumin (500 mg daily for 8 weeks) showed a significant decrease in insulin resistance, ESR, CRP, triglycerides, weight, body mass index, and the waist circumference of RA patients ($p < 0.05$) [97].

Associated with a human cell line study, 3′3-diindolylmethane inhibited proliferation, migration, and the invasion of RA fibroblast-like synoviocytes in vitro and significantly

decreased TNF-alpha-induced increases in the mRNA levels of MMP-2, MMP-3, MMP-8, and MMP-9, as well as the proinflammatory cytokines, such as IL-6, IL-8, and IL-1b [98].

In addition, polyphenolic extract from extra virgin olive oil inhibited IL-1b-induced MMPs, TNF-a, and IL-6 production ($p < 0.001$). IL-1b-induced MAPK phosphorylation and nuclear factor kappa B (NF-kB) activation were also significantly decreased ($p < 0.001$) [99].

Garlic (1000 mg daily for 8 weeks) significantly decreased CRP, TNF-alpha, swollen joint count, pain intensity, tender joint count, DAS-28, and fatigue [100].

Pomegranate extract (500 mg daily for 8 weeks) showed a significant decrease in DAS-28 ($p < 0.001$), pain intensity ($p = 0.03$), and health assessment score (HAQ) ($p = 0.007$). Moreover, glutathione peroxidase concentrations significantly increased ($p < 0.001$), but resulted in no significant difference in MMP-3 and CRP levels between intervention and control groups [101].

Saffron supplementation (100 mg/day for 12 weeks) significantly decreased tender joint count and swollen joint count, pain intensity, and DAS-28. CRP, TNF-alpha, interferon-gamma, and malonedialdehyde (MDA) were also significantly decreased by saffron supplementation [102].

Sesamin supplementation (200 mg/day for 6 weeks) significantly decreased the serum levels of hyaluronidase, MMP-3, CRP, TNF-alpha, and cyclooxygenase-2 and decreased the tender joint count and severity of pain [103].

The inverse relationship with erythrocyte levels of the n-6 PUFA linoleic acid was detected as a risk of RA development (odds ratio 0.29, 95% CI; 0.12–0.75, $p < 0.01$) in patients with RA [104].

Flaxseed supplements (30 g per day for 12 weeks) significantly decreased DAS-28 scores, pain severity, morning stiffness, and feelings of disease, compared to regular diet group [105].

In a prospective case–control study, omega-3 PUFA consumption was inversely significantly associated with omega-6, and the omega-6:omega-3 ratio was directly associated with unacceptable and refractory pain [106]. The dietary recall of average weekly servings of fish had significantly lower DAS-28-CRP scores when compared to RA patients who never ate fish or ate it less than once per month [107]. Fish oil n-3 fatty acids (3 g) and reduced-calorie cranberry juice (500 mL daily) was supplemented as a prospective control study. The fish oil-only group showed improvements in DAS28-CRP and adiponectin, but when consuming fish oil supplements together with cranberry juice, significantly reductions in ESR, CRP, DAS28-CRP, adiponectin, and IL-6 levels compared to controls were observed [108]. Notably, five studies have investigated the role of probiotics. Lactobacillus acidophilus (L. acidophilus), Lactobacillus casei (L. casei), Lactococcus lactis, Bifidobacterium (B.) lactis, and Bifidobacterium (B.) bifidum consumption for 60 days improved inflammatory profiles with reductions in white blood cell count, TNF-alpha, and IL-6 levels, but resulted in no significant difference in IL-10 levels, adiponectin, CRP, ESR, ferritin, or DAS-28 [109].

L. acidophilus, L. casei, and B. bifidum dophilus consumption for 8 weeks decreased serum CRP and improved DAS-28 but resulted in no improvement in oxidative stress levels [110]. Synbiotic supplement intake for 8 weeks significantly reduced CRP, DAS-28 scores, and plasma nitric oxide levels [111]. As seen in an in vitro study, recombinant B. bifidum significantly increased IL-10 levels and inhibited levels of IL-6, IL-8, and TNF-alpha to a higher degree than those from food grade bacteria [112]. B. longum significantly inhibited Th17 cells and IL-17-related genes, as well as several proinflammatory mediators [113]. Regarding herbal studies, Nigella sativa extract (500 mg twice daily for 2 months) significantly decreased CRP levels and DAS28 scores [114]. Stachys schtscheglee-vii tea (2.4 g daily for 8 weeks) markedly reduced DAS-28 scores and serum MMP-3 levels [115]. Xinfeng supplement (three pills, three times a day for 2 months) significantly decreased the DAS-28 score, ESR, and CRP levels [116].

Regarding the effect of anti-oxidants/ROS scavengers, N-acetyl cysteine (600 mg twice daily for 12 weeks) significantly reduced MDA, NO, and total thiol groups compared

to the control [117]. Two other reports for serum silicon levels in patients with RA and vitamin K1 (10 mg daily for 8 weeks) showed controversial results [118,119]. Vitamin A (retinol) detoxifies $H_2O_2$ through ascorbate peroxidase [63]. Vitamin E (tocopherol and tocotrienols) guards against and detoxifies the products of membrane lipid peroxidation. Vitamin C (ascorbic acid) directly scavenges ROS or indirectly supports in the synthesis of extracellular matrix proteins, such as collagen [120]. Selenium exerted antioxidant effects in patients with RA [121]. It is a conjugate factor of selenoenzymes, such as TrxRs and GPx, and showed anti-inflammatory activity by suppressing the NF-κB cascade [122]. Regarding natural products, *Humulus japonicus* (HJ), a widely used herbal medicine in Asia with antioxidative effects, significantly decreased the gross arthritic scores and paw swelling in a collagen-induced arthritic mouse model. HJ also significantly inhibited the expression of IL-6 both in vivo and in vitro. The mechanism underlying HJ effects was regulated by STAT3 phosphoylation [123].

Monotropein, an iridoid glycoside isolated from the roots of *Morinda officinalis* [124], ameliorated $H_2O_2$-induced inflammation in human umbilical vein endothelial cells via NF-κB/AP-1 signaling [125]. In addition, it protected against oxidative stress in osteoblasts via Akt/mTOR-mediated autophagy [126]. Perillyl alcohol is a monoterpene that shows anti-inflammatory and antioxidative properties and can be extracted from widely available essential oils. In RAW264.7 cells, the lipopolysaccharide-induced elevation of IL-1b, IL-6, and TNFα levels was completely inhibited by perillyl alcohol. It also inhibited ROS and nitrite levels via the NF-κB and Nrf2 signaling pathways [127]. Moreover, elastr8ol, a quinone-methylated triterpenoid extracted from *Tripterygium wilfordii*, is used to treat RA. Celastrol-inhibited ROS levels in vitro and attenuated collagen induced arthritis via the NF-κB signaling pathway [128,129]. Genistein, an isoflavone derivative found in soy, decreased the TNFα-induced production of IL-1b, IL-6, and IL-8 in MH7A cells. It also induced NF-κB translocation by TNFα and the phosphorylation of IkB kinase-α/β and IkBα. TNFα-induced adenosine monophosphate-activated protein kinase inhibition was prominently inhibited by genistein [130]. Once IL-21 binds to its receptor, ROS are produced, JAK1/STAT3 signaling is activated, and targeted inflammatory cytokines, such as TNFα, IL-6, MMP-3, and MMP-13, in MH7A fibroblast-like synoviocytes are produced, resulting in the degradation of the extracellular matrix. Nobiletin, a derivative of citrus fruit, attenuated the development of osteoarthritis and inhibited the production of proinflammatory cytokines. Nobiletin potently inhibited the IL21-induced production of ROS by inhibiting the phosphorylation of JAK/STAT3 [131]. Regarding Ayurvedic foods, Gugglipid [132] suppressed collagen degradation and reactive oxygen species in an arthritis mouse model. Kalpaamruthaa also had a suppressive effect on arthritis models [133]. N-acetylcysteine (NAC) is a cysteine prodrug that indirectly activates cysteine-glutamate antiporters [134]. MH7A cells, which are rheumatoid synovial fibroblast-like cell lines, are recombinant cells in which the SV40 T antigen is incorporated into synovial cells collected from RA patients. The amount of ROS produced by MH7A cells was significantly increased at 1 h following the addition of hydrogen peroxide but was significantly decreased following the administration of NAC for 24 h. Therefore, NAC reduced the amount of ROS produced and exerted antioxidant effects in MH7A cells [135]. NAC has antioxidant and detoxifying effects and is a drug clinically prescribed for acetaminophen poisoning and is known to decrease cytokine activity [136]. Nrf2 is the main regulator of the oxidative stress response system [137]. It is normally bound to Keap1; however, when oxidative stress is induced, the conjugated factor p62 is phosphorylated. Nrf2 dissociates from Keap1 and is translocated from the cytoplasm to the nucleus to bind to the oxidative stress response site, promoting the expression of detoxifying enzymes, the antioxidant protein heme oxygenase 1, and anti-inflammatory enzymes. At 24 h after the administration of 1000 μM NAC, the levels of Nrf2 and phosphorylated p62 significantly increased. In a previous clinical study, Nrf2 mRNA and protein levels in the RA synovia were compared to those in the OA synovia. Nrf2 mRNA levels were significantly correlated with the preoperative d-ROM value, suggesting that increased Nrf2 mRNA expression reflects an upregulation of antioxi-

dant capacity in response to high oxidative stresses in RA patients [138]. When cells were observed under a confocal laser scanning microscope, Nrf2, which showed green staining in the untreated cells, was localized in the cytoplasm; however, after NAC administration, Nrf2 was translocated to the nucleus, where it exerted an antioxidant effect. Furthermore, MMP-3 protein levels were significantly reduced by the administration of NAC for 24 h, and JNK phosphorylation was significantly suppressed 3 h after NAC administration. The ROS increase associated with the addition of $H_2O_2$ was significantly reduced by the administration of the JNK inhibitor SP600125 [139], similar to the IL-6 concentration in the MH7A cell supernatant. These findings suggest that the JNK pathway plays an important role in the pathway of oxidative stress and IL-6 suppression [135]. One clinical prospective study showed that 600 mg NAC treatment (twice a day) for 8 weeks significantly improved the RA disease progression and serum IL-17 concentration compared to those of the control group [140].

Conversely, another report showed that NAC oral administration for 12 weeks only had partial effects on the global health parameter, visual analogue scale, and health assessment questionnaire but not DAS28 [141]. Thus, the clinical effect of NAC for the treatment of RA patients remains controversial. Recently, novel thiol-amides, NAC-amide (AD4/NACA), and thioredoxin mimetics (TXM-peptides) have been tested for neurodegenerative disorders [142]. The AD4 compound was effective at blocking cocaine-seeking behaviors [143]. Nevertheless, further in vivo and in vitro investigations of the effect of NAC-amide and thioredoxin mimetics in RA are required. The gut microbiota influences metabolic and immune homeostasis. Oxidative stress, such as the generation of ROS, is the main trigger that directly influences the microbial pattern of human microbiota [144]. Obesity itself is associated with the presence of inflammation, oxidative stress, and mitochondrial dysfunction. Furthermore, these circumstances may develop neurodegenerative diseases, such as Alzheimer disease and Parkinson's disease (PD) [145].

Obesity and neurodegenerative diseases, such as PD, also show dysbiosis (microbiota) [146], and the improvement of dietary pattern (short-chain fatty acid intake) restores this dysregulation pattern [147]. Studies associated with gut microbiota demonstrate that an expansion and/or decrease in bacterial groups is a primary feature in RA compared to the control [148].

The metagenomic shotgun sequencing and a metagenome-wide association study of the fecal, dental, and salivary samples from a cohort of individuals with RA and healthy controls were performed. The results showed that Heaemophilus spp. were depleted and Lactobacillus salivarius was over-represented in patients with RA at all three sites. In very active RA, Lactobacillus salivarius increased [149]. Disease-modifying anti-rheumatic drug (DMARD) treatments partially restore a healthy microbiome because DMARDs improve metagenomic linkage groups (MLGs) in dental and salivary sites after DMARD treatment in patients with RA [149]. Diet is the main environmental factor influencing gut microbiota. The whole dietary pattern of the Mediterranean diet possibly acts as a therapeutic approach by modulating gut microbiota [150]. Red meat and salt are suspected to have harmful effects when controlling the disease activity of RA [151].

In addition, PUFAs, vitamin D, and probiotics supplementation demonstrated protective effects regarding RA development by improving the environment of gut microbiota. Healthy lifestyle and nutrition are encouraged for patients with RA [151].

## 9. Conclusions

We described the role of ROS in the pathogenesis of rheumatoid arthritis. ROS are elevated in the serum of patients with RA. The biomarkers of oxidative stress were demonstrated. Oxidative stress was reported to be associated with autophagy/ER stress in the pathogenesis of RA. In RA, the main regulator of redox signaling is the Nrf2/Keap1 pathway. New therapeutic targets and natural food or phytochemicals with the potential for improving the severity of RA were described. Dietary patterns, Mediterranean diet (MD),

flavonoids, PUFAs, probiotics, herbals, and antioxidants are useful for decreasing RA disease severity.

Recently, there has been progress in the dysregulation of gut microbiota in patients with RA, and the improvement of the environment of gut microbiota by diet would be a target for future research.

**Author Contributions:** Conceptualization, N.K.; methodology, N.K., T.K. and M.O.; validation, N.K.; data curation, N.K., T.K. and M.O.; writing—original draft preparation, writing—Review and editing, supervision, N.K.; visualization; T.K. and M.O. All authors have read and agreed to the published version of the manuscript.

**Funding:** This research received no external funding.

**Institutional Review Board Statement:** The study was conducted in accordance with the Declaration of Helsinki and approved by the Institutional Review Board of Niigata University (protocol code: 2018-0377 and date of approval: 25 July 2019).

**Informed Consent Statement:** Informed consent was obtained from all subjects involved in the study.

**Data Availability Statement:** Not applicable.

**Acknowledgments:** The authors would like to acknowledge the support of Rika Kakutani (Division of Orthopedic Surgery, Niigata University) in collecting and reviewing the reference articles and drawing Figure 1, and Akiko Yoshi (Department of Neurosurgery, Brain Research Institute, Niigata University) for providing exclusive support with the experiments.

**Conflicts of Interest:** The authors declare no conflict of interest.

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
