# Peer review of "Rheumatoid Arthritis and Reactive Oxygen Species: A Review"

_cimb, doi:10.3390/cimb45040197_

Round 1
Reviewer 1 Report
Comments and Suggestions for Authors
The authors present a review of ROS and RA and possible treatment with antioxidants.
Only facts are presented without expressing a personal opinion and no future trends are delineated.
Lines 63-64 - Methotrexate is not a TNF inhibitor
Line 114-115 - miR-126 could not be a candidate for the treatment of RA, instead, it could be a predictive biomarker
Lines 141-146 are copy-paste from Ref. 69, however, it ends with a citation of Ref. 70.
Abbreviations should be checked in the entire document - some of them appear only once and are unnecessary to be given (for example - CML, IsoPs, TBARS).
Table 1 should be entirely revised – What is the meaning of Table 1?; Are the cited references in Table 1 really proper? Does a single case report the most proper reference or it is better to mention a review?
Lines 241-245 are inappropriately presented as the method is already published by the authors.
Line 189-201 - The text should be entirely revised.
Line 281 - the cited Ref. needs to be 117.
The conclusion has no sense of conclusion. Lines 285-292 should delineate the future trends and present the opinion of the authors concerning RA, ROS, and treatment with antioxidants. In fact, the meaning of lines 285-292 does not differ from the aim of this review (Lines 50-54) at the end of the Introduction. It is unclear why the authors conclude that isoprostane is the most promising biomarker since this statement is emphasized nowhere in the text.
Author Response
Please see the attachment. Naoki Kondo

Reviewer 2 Report
Comments and Suggestions for Authors
Dear authors,
After the review process, I have several comments: you do not present any data related to the human microbiota, which is an important aspect in the pathogenesis of rheumatoid arthritis; in section 8, you should include details related to the bioactive compounds from natural sources and microbiota pattern; you should include comments that prove the correlations between microbiota bioactivity and bioavailability of functional compounds - data relevant for you in the case of rheumatoid arthritis development and progression; a link between obesity, microbiota dysbiosis, and other degenerative diseases is essential as future perspectives of the paper.
Best regards!
Author Response
Dear, Dr.Maxima Mei, Associate Editor of Current Immunology of Molecular biology (CIMB) and the reviewers,
I appreciate that you reviewed our manuscript. According to your reviewed comments, I (the first author and the corresponding author), Naoki Kondo, revised the manuscript (ID2241698) and I would like to respond to the comments you mentioned one by one.
Reviewer 2
Dear authors,
After the review process, I have several comments: you do not present any data related to the human microbiota, which is an important aspect in the pathogenesis of rheumatoid arthritis; in section 8, you should include details related to the bioactive compounds from natural sources and microbiota pattern; you should include comments that prove the correlations between microbiota bioactivity and bioavailability of functional compounds - data relevant for you in the case of rheumatoid arthritis development and progression; a link between obesity, microbiota dysbiosis, and other degenerative diseases is essential as future perspectives of the paper.
Best regards!
→ Thank you so much for the precious comment.
Actually, the association between dysbiosis and RA pathogenesis is important and relevant content in our manuscript. In the last of Section 8, I added as follows;
“The gut microbiota influences metabolic and immune homeostasis. Studies associated with gut microbiota demonstrate that an expansion and/or decrease in bacterial groups is one of feature in RA compared to control [144].
The metagenomic shotgun sequencing and a metagenome-wide association study of the fecal, dental, and salivary samples from a cohort of individuals with RA and healthy controls were performed. The results showed that Heaemophilus spp. were depleted and Lactobacillus salivarius was over-represented in patients with RA at all three sites. In very active RA, Lactobacillus salivarius increased [145].
Disease modifying anti-rheumatic drugs (DMARDs) treatment partially restores a healthy microbiome because DMARDs improved metagenomic linkage groups (MLGs) in dental and salivary sites after DMARDs treatment in patients with RA [145].
Diet is the main environmental factor influencing gut microbiota. Mediterranean diet, whole dietary pattern possibly act for therapeutic approach by modulating gut microbiota [146]. Red meat and salt have been suspected harmful effect for controlling disease activity of RA [147].
In addition, PUFAs, vitamin D, and probiotics supplementation showed protective effect for RA development by improving the environment of gut microbiota. Healthy lifestyle and nutrition are encouraged for patients with RA [147]. “.
Please check and review again.
Sincerely yours,
3-18-2023
Division of Orthopedic Surgery, Niigata University Graduate School of Medical and Dental Sciences, Niigata, Japan.
Naoki Kondo

Round 2
Reviewer 1 Report
Comments and Suggestions for Authors
Dear Authors,
The text from line 193 to line 199 must be removed.
It is meaningless! Just delete it, please.
Author Response
Reviewer 1
“The text from line 193 to line 199 must be removed. It is meaningless! Just delete it, please.”.
→ Thank you so much for the comment, As you mentioned above, I deleted the text from 193 to 199. 
Please review the re-revised manuscript again.
King regards,
3-26-2023, a corresponding author
Naoki Kondo
Reviewer 2 Report
Comments and Suggestions for Authors
Dear authors,
You should expand in deep my comment related to the link between obesity, microbiota dysbiosis, and other degenerative diseases is essential as future perspectives of the paper.
Best regards!
Author Response
Reviewer 2
Comment;
“You should expand in deep my comment related to the link between obesity, microbiota dysbiosis, and other degenerative diseases is essential as future perspectives of the paper.”.
→ Thank you so much for the precious comment.
I added the sentences related obesity, microbiota dysbiosis, and other degenerative diseases and also added the citations as following;
“Oxidative stress such as the generation of ROS is the main triggers that directly influences the microbial pattern of human microbiota [144]. Obesity itself is associated with the presence of inflammation, oxidative stress, and mitochondrial dysfunction. Furthermore, these circumstances may develop neurodegenerative diseases such as Alzheimer disease and Parkinson’s disease (PD) [145]. Obesity, neurodegenerative disease such as PD also show dysbiosis (microbiota) [146] and improvement of dietary pattern (short-chain fatty acid intake) restores this dysregulation pattern [147].”.
* Cited references
- Vamanu, E. Poluphenolic nutraceuticals to combat oxidative stress through microbiota modulation. Front. Pharmacol. 2019, 10, 492. DOI:10.3389/fphar.2019.00492.
- Mazon, J.N.; de Mello, A.H.; Ferreira, G.K.; Rezin, G.T. The impact of obesity on neurodegenerative diseases. Life. Sci. 2017, 182, 22-28. DOI: 10.1016/j.lfs.2017.06.002.
- Vamanu, E.; Rai, S.N. The link between obesity, microbiota dysbiosis, and neurodegenerative pathogenesis. Diseases, 2021, 9, 45. DOI; 10.3390/diseases/903045.
- Uyar, G. Ö.; Yildiran, H. A nutritional approach to microbiota in Parkinson’s disease. Biosci. Microbiota Food and Health, 2019, 38, 115-127.
Accompanied to these contents, I changed the numbering of the citations from the original 144 through 147 to the revised 148 through 151.
Please review the re-revised manuscript again.
King regards,
3-26-2023, the corresponding author
Naoki Kondo